# Using Generative Models for Pediatric wbMRI

**Alex Chang**[*1,2,3]                                                  CHANGA47@CS.TORONTO.EDU

**Vinith M. Suriyakumar**[*1,2,3]                                          VINITH@CS.TORONTO.EDU

**Abhishek Moturu**[*1,2,3]                                            MOTURUAB@CS.TORONTO.EDU

**Nipaporn Tewattanarat**[3]                        NIPAPORN.TEWATTANARAT@SICKKIDS.CA

**Andrea Doria**[3]                                             ANDREA.DORIA@SICKKIDS.CA

**Anna Goldenberg**[1,2,3]                                ANNA.GOLDENBERG@UTORONTO.CA

[1] *Department of Computer Science, University of Toronto, Toronto, Canada*

[2] *Vector Institute, University of Toronto, Toronto, Canada*

[3] *The Hospital for Sick Children, Toronto, Canada*

## Abstract

Early detection of cancer is key to a good prognosis and requires frequent testing, especially in pediatrics. Whole-body magnetic resonance imaging (wbMRI) is an essential part of several well-established screening protocols, with screening starting in early childhood. To date, machine learning (ML) has been used on wbMRI images to stage adult cancer patients. It is not possible to use such tools in pediatrics due to the changing bone signal throughout growth, the difficulty of obtaining these images in young children due to movement and limited compliance, and the rarity of positive cases. We evaluate the quality of wbMRI images generated using generative adversarial networks (GANs) trained on wbMRI data from The Hospital for Sick Children in Toronto. We use the Fréchet Inception Distance (FID) metric, Domain Fréchet Distance (DFD), and blind tests with a radiology fellow for evaluation. We demonstrate that StyleGAN2 provides the best performance in generating wbMRI images with respect to all three metrics.

**Keywords:** machine learning, generative models, cancer detection, MRI, whole body MRI

## 1. Introduction

Whole-body magnetic resonance imaging (wbMRI) is an essential part of well-established cancer screening protocols (Villani et al., 2016). These protocols were shown to improve early detection of cancer for both adult (Attariwala and Picker, 2013) and pediatric (Greer et al., 2017) patients. Machine learning methods have been successfully applied in staging adult cancer patients from wbMRIs (Lavdas et al., 2019). The same task is much more challenging for pediatric patients due to i) varying bone signals during growth, ii) the movement and limited compliance of young children during imaging, and iii) the rarity of positive cases. The lack of training data suggests the need for alternatives to standard CNN-based (Bien et al., 2018) approaches or augmentation-based detectors.

Generative models, such as generative adversarial networks (GANs), have shown promise in anomaly detection in numerous medical imaging applications (Yi et al., 2019). Given the

---

[*] Contributed equally

need for an automated pediatric wbMRI cancer screening tool, we set out to study different generative models for the primary task of generating pediatric wbMRIs. We limited our study to evaluating the generation of images. The quality of the generated images can be seen as a measure of how well the model has captured the underlying data distribution which is essential to the eventual downstream task of cancer screening (anomaly detection). We applied these models to 360 wbMRI slices from The Hospital for Sick Children in Toronto. We trained multiple GAN architectures and used Fréchet Inception Distance (FID), Domain Fréchet Distance (DFD), and radiology blind tests to evaluate the image quality of each model.

We demonstrate that StyleGAN2 generates the best quality images and that DFD is a promising metric to compare image quality. We also demonstrate our preliminary results on the task of anomaly detection. Our analyses characterize the use of generative models for medical image generation and potential downstream tasks such as anomaly (cancer) detection, contributing to the much needed advances in pediatric medical imaging.

## 2. Methods

Our dataset is comprised of 90 de-identified healthy patients from a pediatric hospital, including males and females of ages 4 to 18. Four middle anatomically similar slides were selected from each volume. Each slice was preprocessed using N4ITK bias field correction (Tustison et al., 2010), contrast-limited adaptive equalization (Kaur and Rani, 2016), and noise reduction (Senthilkumaran and Thimmiaraja, 2014). We cropped and padded the images to be a uniform size of $800 \times 256$ to register the position of different patients.

We trained (Appendix A) four different generative models: DCGAN (Radford et al., 2015), StyleGAN (Karras et al., 2018), StyleGAN with progressive training (PGStyleGAN) (Karras et al., 2017), and StyleGAN2 (Karras et al., 2019). For evaluation, we measured the FID (Heusel et al., 2017) and the DFD in the feature space of a Variational Autoencoder (VAE) trained on the same dataset according to (Liu et al., 2018). For our blind tests with our radiology fellow, we randomly chose 10 real images and 10 generated images from each model. We then showed the radiologist each of the images in random order asking them to classify the image as real or fake (generated).

Finally, we performed anomaly detection using a GAN trained with healthy images (Schlegl et al., 2017). With a query image, we find the closest generated image and subtract the two images to provide areas of high disease probability Figure 2. Cancer tumours are simulated by generating a set of circles around a point on the image with varying pixel intensities and radii. For future work, we are working on acquiring and using real cancer images instead of simulating tumours. We compared the accuracy of our anomaly detection to watershed segmentation (Mustaqeem et al., 2012) which is traditionally used in low data settings as its performance is agnostic to data amount. This method is not the state of the art in classical image segmentation but it is a commonly used method that is low resource and fast which is why we selected it.

## 3. Results

**Generated Image Quality.** Figure 1 shows samples from StyleGAN2 have the highest visual quality, which is supported by the error rate in classification by our radiologist in Table 1. The radiologist was able to detect most images were fake across all of the chosen architectures most commonly due to artifacts generated by the model which would not be present in real images. Furthermore, we observed that StyleGAN2 generates more diverse samples and does not suffer as much from mode collapse compared to other approaches.

**Domain Fréchet Distance Metric.** We observed that the FID metric is inconsistent with the visual quality of samples for this domain since StyleGAN2 should have the lowest FID (see Table 1 and Figure 1). We hypothesize the reason to be that our wbMRI images are very different from natural images used to train Inception v3. The DFD in the VAE feature space successfully captures the order of model performance for the same dataset.

**Anomaly detection.** Figure 2A shows a proof-of-concept of the anomaly detection method proposed by (Schlegl et al., 2017) for wbMRI. Since the GAN is only trained using healthy images, by finding the closest image in the generative distribution, we can highlight anomalous areas in a diseased query. We demonstrate that our GAN outperforms the classic watershed segmentation in Figure 2B.

## 4. Conclusion

In this paper, we demonstrate that state-of-the-art GANs are able to generate pediatric wbMRIs needed to enable automated cancer detection. In particular, samples generated using the StyleGAN2 architecture had high enough visual fidelity that our radiologist classified them as real. We also demonstrate that the FID metric used in the GAN literature is inappropriate for this domain and that DFD is a promising alternative. Finally, we show a downstream task of anomaly detection, using the GAN trained on healthy images to detect cancerous lesions, which may mitigate the need for scarce examples of wbMRIs with cancer.

Table 1: FID and DFD scores (with VAE Features) along with the false positive rate for the radiologist blind test for each of the GAN architectures.

| Model | FID | DFD | Radiologist False Positive Rate |
|---|---|---|---|
| DCGAN | 457.30 | 23.72 | 0% |
| StyleGAN | 481.3 | 19.378 | 0% |
| PGStyleGAN | 442.61 | 18.56 | 20% |
| StyleGAN2 | 497.09 | 17.234 | 30% |

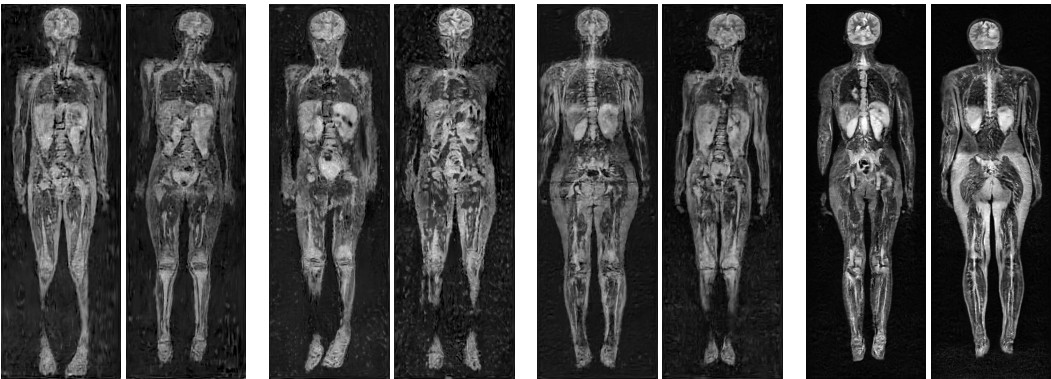

Figure 1: From left to right, two images generated by each of the following GAN architectures: DCGAN, StyleGAN, PGStyleGAN, and StyleGAN2.

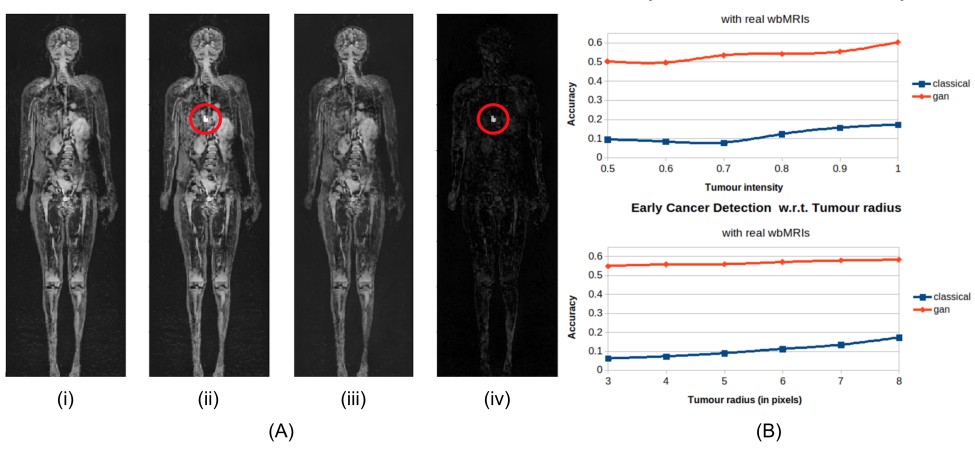

Figure 2: (A): (i) query wbMRI; (ii) with imputed tumour; (iii) the nearest image in the generator's modeled distribution; (iv) the absolute difference between (ii) and (iii). (B): Accuracy of our GAN anomaly detection vs the watershed segmentation as a function of simulated tumours' pixel intensities (top) and tumour radii (bottom).

## Acknowledgments

We acknowledge the support of the Natural Sciences and Engineering Research Council of Canada (NSERC) and the The Mark Foundation for Cancer Research. Resources used in preparing this research were provided, in part, by the Province of Ontario, the Government of Canada through CIFAR, and companies sponsoring the Vector Institute www.vectorinstitute.ai/#partners.

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

## Appendix A. GAN Training Settings

| Model | $\alpha$ | Batch Size | Instance Noise steps |
|---|---|---|---|
| DCGAN | 0.001 | 30 | 10K |
| StyleGAN | 0.001 | 12 | 10K |
| PGStyleGAN | 0.001 | 360,360,180,60,30,15* | 10K** |
| StyleGAN2 | 0.002 | 12 | 0 |
| VAE | 0.001 | 45 | N/A |

Table 2: Hyperparameters used during GAN training

* Batch size at each progressive growth step between 1 and 6, respectively.
** After the complete growing of the last layer.

In all models, a similar architecture skeleton is used to upsample noise (512) to the image resolution ($800 \times 256$). The Generator first upsamples noise to 512 feature maps of size $25 \times 8$ with a fully connected layer. Next, 5 convolutional blocks, each consisting of two convolutional layers (with $3 \times 3$ filters and stride 1) and an upsampling layer (by bilinear interpolation) in between, are used to double the width and height of the feature maps. The number of feature maps are also halved in the last 3 blocks. The result of dimension $64 \times 800 \times 16$ is passed to one last convolutional layer to obtain the grayscale image. The discriminator is almost a mirror of the generator; it obtains intermediate feature maps of the same dimension with similar convolutional blocks, but downsamples the width and height with a convolutional stride of 2. Two fully connected layers of size 512 and an output layer are added at the end. The remaining hyperparameters, and training details are inherited from the original StyleGAN paper.

For the training of DCGAN, StyleGAN and PGStyleGAN, Gaussian noise with $\sigma = 0.2$ is independently added to each pixel in both real and fake images and $\sigma$ is linearly reduced to 0 in the number of steps indicated in Table 2. In the training of all models, a latent dimension size of 512 is used to sample Gaussian noise.

