# OpenReview forum: "Using Generative Models for Pediatric wbMRI"
_MIDL.io/2020/Conference — MIDL 2020_

### Official Review · AnonReviewer4 · 2020-03-11
**A paper comparing several GAN methods to generate wbMRI images**

**Rating:** 3
**Confidence:** 3

**Review:**

This paper compares several GAN methods in terms of generating paediatric wbMRI images. They compared different GANs with two metrics commonly used in computer vision and a real v.s. fake human test. They also used the generated images for cancer detection task with comparison with a classical method.
Pros:
 - Compare several GAN methods including some very recent ones.
- Use metrics to evaluate results including a human test.
- Showed convincing qualitative results.

Cons:
- FID and DFD may not be suitable to evaluate medical images. One key point of medical image synthesis is that synthesised images do not only need to be variant and realistic, but need to be clinically meaningful.
- Not clear what is the input of the GANs? Is it a random vector/scalar? Or is it a real medical image? If you want to perform detection by comparing generated images with real images, is it better to use real medical images as input, instead of finding closet neighbour?
- What are the data you used? Are they publicly available?
- Is the classical method you compared with the state-of-the-art? Or how far it is from the state-of-the-art? It seems that its results are quite poor.

---

### Official Review · AnonReviewer3 · 2020-03-12
**Review 3**

**Rating:** 2
**Confidence:** 5

**Review:**

The paper evaluates pediatric whole MRI generation using 4 pre-existing GAN models.
The evaluation consists of qualitative visualization, as well as FID, DFD and radiologist discriminative rate for real/fake images. They also conduct a synthetic anomaly detection task (i.e. imputing the healthy MRI with an artificial anomaly) and show the model being able to identify the inserted artifact.

I have some major concerns which I will list:
1) The research question behind this work is not clear to me. In the introduction, the authors state the motivation for this work is to develop a cancer screening tool. However, the experiments do not reflect that. If the research question is how GANs can be used for cancer screening in wbMRI, then they should have evaluated the model on a real anomaly detection task, not a synthetic one.

2) In the synthetic anomaly detection task, it is not clear if the query image is from the training data or a held-out test data. Based on what I infer from the paper, it seems the authors used the whole 90 subjects for training. which implies the query image was from the training data. This impairs the validity of the experiment since the generator could have over-fitted to the training data.  I would like to see a held-out dataset consisting of multiple subjects.

3) There is no qualitative or quantitative measure to show if the generative model has overfitted to the training data. One way to show this is to show for every generated image the closest neighbor in the training data. While this is not a quantitative measure, it is a qualitative one. Also what metric to use to find the nearest neighbor could be tricky. You could use the same metric you used in the anomaly detection task.

I think the paper needs more experiments to validate the approach and so I would reject it in the current state.

---

### Official Review · AnonReviewer2 · 2020-03-12
**An analysis of different (Style)GAN variants for synthesis of high resolution wbMRI with an anomaly detection use-case experiment**

**Rating:** 4
**Confidence:** 5

**Review:**

The paper is well written and interesting to read. The experimental setup is made very clear and results are nicely portrayed, alone the motivation for wbMRI synthesis does not convey completely from the very beginning. I really appreciate the combination of PGAN/StyleGAN and Schlegl's anomaly detection approach. I also really like the study of simulated tumor intensity, radius and the impact on anomaly detection, although these preliminary results are not earth-shattering.


Pros:
- well written and clearly motivated

Cons:
- Given the rare nature of cancer in pediatrics, can you comment on the clinical relevance of this field of study? Do you see potential elsewhere?
- Cancer regions are only simulated, a set of real cancer testing data would have been nice (is such data available? Again, I think this relates to the clinical relevance)
- Anomaly detection: which model did you exactly use for anomaly detection?
- DCGAN: I really wonder how you were able to obtain such compelling results using DCGAN. It would be great if you could provide details on the training
- Anomaly detection: Do you also have visual results of a less hyper-intense, simulated tumor?
- Anomaly detection: Is the accuracy evaluated on a pixel-level, or on a level of connected components?
- Where exactly did the StyleGAN2 fail, as the radiologist was still able to correctly identify 70% of the generated images as fake.

Minor:
- Introduction: How the synthesis of such wbMRI would play together with anomaly detection is not completely obvious from the very beginning, I'd suggest some rephrasing for the introduction.

---

### Meta-Review · Area_Chair1 · 2020-04-06
**MetaReview of Paper312 by AreaChair1**

**Rating:** 3

**Metareview:**

This paper compares several GAN methods for generating pediatric wbMRI images.

The paper is well written and results although limited are clear and interesting.

The reviewers see merits in such a short paper and mostly think that it is worth to be presented at MIDL.



**Paper Type:**

validation/application paper

---

### Decision · Program_Chairs · 2020-04-11

Accept